# Impacts of Seasonal and Annual Weather Variations on Network-Level Pavement Performance

**Lu Gao [1], Feng Hong [2] and Yi-Hao (Wilson) Ren [1,\*]**

[1] Department of Construction Management, University of Houston, Houston, TX 77204, USA; lgao5@central.uh.edu

[2] Texas Department of Transportation, Austin, TX 78731, USA; feng.hong@txdot.gov

[\*] Correspondence: yren10@uh.edu; Tel.: +1-(832)-298-7825

**Abstract:** This study was aimed to determine the extent of association between network-level pavement condition and seasonal and annual weather variations. Pavement condition data recorded in the Texas Department of Transportation's Pavement Management Information System (PMIS) database between 2000 and 2008 was highlighted. Meteorological data collected at different districts across the whole state were used in the explanatory variables. Dynamic panel data analysis was used in the deterioration models to quantify the effects of temperature and precipitation variations on pavement conditions. Based on the statistical implication from the model estimation results, significant correlations were identified between pavement conditions, the average monthly rainfall, and the average monthly temperature recorded 1 to 23 months prior to pavement condition inspection.

**Keywords:** pavement performance; seasonal effect; panel data; pavement management; PMIS

## 1. Introduction

The climatic factors affecting pavements are mainly temperature and precipitation. While temperature affects the resilient modulus of asphalt material and causes curling of concrete slab, moisture entering the pavement structure through cracks may reduce the strength of pavement and subgrade [1]. In the past, studies have been conducted to quantify the effects of climatic variations on pavement performance at the project level and from a pavement design viewpoint, which usually has adopted an empirical–mechanistic approach. For example, Ongel and Harvey [2] evaluated the effects of temperature and rainfall on pavement distress across the climate regions in California. The authors used the Enhanced Integrated Climatic Model (EICM) to simulate hourly pavement temperature at the critical depths in the pavement layers. The effects of temperature and precipitation on rutting, cracking, and faulting were then compared for different regions. Tighe et al. [3] and Mills et al. [4] use Canadian data from the Long-Term Pavement Performance (LTPP) database and the Mechanistic Empirical Pavement Design Guide (MEPDG) to quantify the impact of projected climate factors on pavement performance of low-level volume roads. Pavement roughness, longitudinal cracking, alligator cracking, transverse cracking, and deformation under different climate scenarios were estimated. Meagher et al. [5] use climate model data sets at four different sites across New England as inputs into the MEPDG model, in order to simulate flexible pavement performance and deterioration over time. The authors found that while the simulated impact of climate changes is negligible for alligator cracking, the asphalt pavement rutting differences were significant.

The effects of climatic factors on network-level pavement performance have also been studied extensively by previous researchers. Mohamed and Hansen [6] use wheel loading temperature and moisture gradients as the affecting factors to study tensile stress at both the top and bottom of a slab. The first part of the study presented the calculation of self-equilibrated stresses caused by internal

restraint within a cross-section. The second part of the study calculated stresses caused by external restraint, using an equivalent linear temperature gradient provided by the first part of the study. Bosscher et al. [7] develop a statistical model to analyze the pavement temperature using meteorological data. The study used air temperature, relative humidity, wind speed and direction, and solar radiation as the affecting variables. The developed model was compared with the Superpave model and the model recommended by the LTPP program. The results of this study indicate that at air temperatures higher than 30 °C, the high pavement design temperature was underestimated by the Superpave and LTPP models. The study also found significant differences between the standard deviation of air and pavement temperature. Yu et al. [8] utilize both temperature and wheel loads as the affecting variables and studied concrete pavement responses to those affecting variables. It was discovered that the built-in upward curling of a concrete slab shares a similar effect as negative temperature gradients. Bäckström [9] utilize both temperature and precipitation as the affecting variables, and it was found that a porous pavement is more resistant to freezing than traditional impermeable pavement, due to higher latent ground heat caused by higher water composition in the underlying soil. In Marasteanu et al. [10], two models are developed to study the cracking mechanism of asphalt pavement exposed to low temperature. The first model used temperature as the affecting variable to predict the crack spacing in asphalt pavement exposed to low temperature. The second model also used temperature as the affecting variable, but to predict cracking propagation and accumulated damage. The result shows that after a cracking start, it only takes around a 0.5 °C decrease in temperature for the crack to propagate. In Bazlamit and Reza [11], temperature and surface texture are used as the affecting variables to study the friction force developed during skidding. Ten field sites with different pavement types and five different temperatures were used in this study. The result shows that the hysteresis component of friction decreases as temperature increases, regardless of the pavement texture state. The adhesion component was more sensitive to the effect of surface texture. Econometric models were also used in these works. For example, Loizos and Karlaftis [12] use a duration model to predict pavement failure times, based on data from Europe's road network. In this study, average daily temperature, number of days per year with a maximum temperature above 25 °C, number of days per year with a minimum temperature below 0 °C, freezing index, and average yearly precipitation were used as explanatory variables. Madanat et al. [13] develop performance models using data from Washington State's Pavement Management System (PMS) databases to predict the initiation of cracking and progression of roughness in both asphalt and Portland cement pavements. In this research, climate-related explanatory variables include the maximum temperature during the hottest month, the minimum temperature during the coldest month, the number of freeze–thaw cycles, and the average annual precipitation. Hong and Prozzi [14] develop a Bayesian-based incremental pavement performance model using experimental data from the American Association of State Highway Officials (AASHO) road test. A frost penetration gradient was used as an environmental factor in the model. Nakat and Madanat [15] developed a pavement overlay crack initiation model, using condition survey data for the highway system in the state of Washington. The authors used a semiparametric Cox model to capture the main factors responsible for the overlay crack initiation process. In this model, the average monthly minimum temperature of the coldest month and the annual precipitation were used as climate-related explanatory variables. The authors found that as the minimum temperature increases, the occurrence of low-temperature cracking in the pavement will decrease. Yu et al. [16] used a pavement condition rating (PCR) as a performance indicator, and developed a survival model to predict the failure times of pavement. The survival data of asphalt overlays on flexible pavements in Ohio were used in this study. Regional annual average snowfall, regional annual average temperature, and regional annual average precipitation were used as the explanatory variables for the survival model. The authors found that a 1 °C increase of the temperature leads to about a one year increase in the service life of the pavement, and every 10 cm increase in precipitation leads to a one-year decrease in service life. Hong and Prozzi [17] use in-service pavement sections data from the Minnesota Road Test project to capture pavement deterioration process. Frost heave was used as an environmental

variable. Ker and Lee [18] apply linear mixed-effect models to the AASHO road test data to predict the flexible pavement serviceability index. Climatic variables, including monthly humidity, precipitation, temperature, freezing index, freeze–thaw cycles, snowfall, days above 32 °C, and days below 0 °C were used in the model. Gao et al. [19] developed a frailty model to study heterogeneity in pavement fatigue cracking. The authors used the average total annual precipitation, average total annual snowfall, average number of days above 89.6 °F, and average freezing index as environmental variables. In Qiao et al. [20], climate factors, such as temperature variation, precipitation, wind speed, and ground water level are used as affecting variables to examine pavement deterioration and service life through sensitivity analysis. It was concluded that both annual temperatures increase, and seasonal temperature variation has the most significant effect on pavement deterioration rate and pavement service life. Reger et al. [21] develop a pavement-crack-initiation model, using the AASHO road test data and the field data from the Washington State Department of Transportation. Annual precipitation and average monthly minimum temperature of the coldest month were used as the climatic factors in the model. Wang [22] develops an ordinal logistic regression model to predict the probability of severity levels for alligator cracking. Freeze–thaw cycles and the amount of precipitation per wet day were used as explanatory variable. Dong and Huang [23] use a survival model with Weibull hazard function to evaluate the influence of different factors on the crack initiation of resurfaced asphalt pavement. Data from Specific Pavement Study (SPS-5) experiments of the LTPP program were used. The authors found that severe freeze–thaw conditions accelerated the occurrence of non–wheel path longitudinal and transverse cracks. Lethanh et al. [24] model the statistical interdependency between different pavement distresses' deterioration process using a Poisson hidden-Markov model. The model describes the complex process of pavement deterioration, which includes the frequent occurrence of local damage (e.g., potholes) and the degradation of other pavement indicators (e.g., cracks, roughness). The hazard rate of the model can be expressed as a function of characteristic variables, including the ambient temperature. Anastasopoulos and Mannering [25] use seemingly unrelated regression equations and duration model to identify influential factors affecting pavement service life. The average annual range of temperature and average annual precipitation were used as the climatic factors. Mohd Hasan et al. [26] use mean temperature and precipitation as the affecting factors to study the effect on the distress of flexible pavement. Seventy-six pavement sections from 13 states with different weather conditions were utilized in this study. The results show that longitudinal cracking of flexible pavement was significantly affected by both temperature and precipitation, whereas alligator cracking and transverse cracking on permanent deformation were only affected significantly by mean annual temperature. However, both temperature and precipitation did not show significant effects on the international roughness of flexible pavement.

Even though some climatic variables were used as explanatory variables in the existing works discussed above, seasonal weather information was not systematically analyzed. This is because, in these cases, the average temperature and precipitation values were usually calculated over a period of many years. There is little research on the effects of seasonal and annual weather variations on network-level pavement performance. For this reason, in this paper, a deterioration model is developed to capture the scale of deterioration occurring as a result of seasonal and annual weather variations. This model was tested on data from the Texas Department of Transportation (TxDOT)'s Pavement Management Information System (PMIS).

## 2. Methodology

In the present study, the dynamic model described by Equations (1)–(6) is used; it has proved its effectiveness in the previous several research studies (Wang et al. [27]; Lee [28]; Wang et al. [29]; Lee and Russell [30]; Chu and Durango [31]). Wang et al. [27] proposed an unevenly spaced dynamic panel model using a dataset collected from the fifth ring road in Beijing to study the seasonal pattern of the ride quality index. The research concluded that the ride quality index collected during autumn is expected to be lower than that of spring and summer. Lee [28] developed a dynamic prediction

model, using panel data to predict highway pavement contractors' performance based on their past performance. The model adopted a random effect model to account for random variables, and was solved by Statistical Analysis System (SAS) software. The result shows that the best model developed in the research has an average Mean Absolute Percentage Error (MAPE) of 16%, and asphaltic concrete pavement construction quality measured in International Roughness Index (IRI) could be predicted based on a contractor's past performance. Wang et al. [29] developed another dynamic panel model by using Asphalt Concrete overlay thickness, average daily traffic (ADT), and Portland Cement Concrete condition before fracture as potential performance influencing factors; however, due to the unavailability of climate data, the influence of climate was treated as uniform. The result of this model shows an acceptable prediction error of 15% compared to actual value. Lee and Russell [32] studied the relationship between the as-built IRI of asphaltic concrete pavement and construction characteristics using panel data. This research shows that a dynamic panel model that accounts for random effects provides the best prediction accuracy. The research also shows that pavement's length and depth is significantly and negatively related to the pavement's as-built IRI—however, the as-built roughness is not strongly affected by the pavement type and volume. Chu and Durango [31] developed a dynamic panel model by using time series models as a framework. The framework provided the dynamic panel model with the ability to capture the serial dependence and exogenous factors effect. The research utilized 308 road sections from the AASHO road test, and the model developed has a root mean square error of 0.56 for out-of-sample predictions, which outperformed other models studied in the research.

The dynamic panel model utilized in the research has the following general form:

$$y_{it} = \rho y_{i,t-1} + x'_{it}\beta + \gamma t + \alpha_i + \epsilon_{it}, \ i = 1, \ldots, N, \ t = 1, \ldots, T \tag{1}$$

where $y_{it}$ is the pavement performance indicator, subscript $i = 1, \ldots, N$ is the section, and $t = 1, \ldots, T$ is the inspection time. The lagged value $y_{i,t-1}$ on the right-hand side of the equation is included to capture the dynamics in the pavement deterioration process. The vector $x_{it}$ accounts for covariates (e.g., temperature and precipitation) that have effects on the deterioration of pavement conditions. The term $t$ captures the time effect, and $\alpha_i$ allows for an unobserved specific factor of the $i$th pavement section. In addition, $\epsilon_{it}$ is the error term, and $\beta$ and $\gamma$ are regression coefficients. The above equation can be represented in matrix form as

$$Y = \rho Y_{-1} + X\beta + \gamma W + D\alpha + \epsilon \tag{2}$$

where $Y = \begin{bmatrix} Y_1 \\ \vdots \\ Y_n \end{bmatrix}_{n \times T}$, $Y_i = \begin{bmatrix} y_{i1} \\ \vdots \\ y_{iT} \end{bmatrix}_{T \times 1}$, $X = \begin{bmatrix} X_1 \\ \vdots \\ X_n \end{bmatrix}_{n \times T, m}$, $X_i = \begin{bmatrix} X'_{i1} \\ \vdots \\ X'_{iT} \end{bmatrix}_{T \times m}$, $\beta = \begin{bmatrix} \beta_1 \\ \vdots \\ \beta_m \end{bmatrix}_{m \times 1}$,

$Y_{-1} = \begin{bmatrix} Y_{1,-1} \\ \vdots \\ Y_{n,-1} \end{bmatrix}_{n \times (T-1)}$, $Y_{i,-1} = \begin{bmatrix} y_{i1} \\ \vdots \\ y_{i,T-1} \end{bmatrix}_{(T-1) \times 1}$, $W = \begin{bmatrix} w_1 \\ \vdots \\ w_n \end{bmatrix}_{n \times T}$, $w_i = \begin{bmatrix} 1 \\ 2 \\ \vdots \\ T \end{bmatrix}_{T \times 1}$, $D = I_n \otimes 1_T =$

$\begin{bmatrix} 1_T & 0 & 0 \\ 0 & \ddots & 0 \\ 0 & 0 & 1_T \end{bmatrix}_{n \times T}$, and $1_T = \begin{bmatrix} 1 \\ \vdots \\ 1 \end{bmatrix}_{T \times 1}$.

Equation (3) is derived by taking the first difference on both sides of the equation

$$y_{it} - y_{i,t-1} = \rho(y_{i,t-1} - y_{i,t-2}) + (x_{it} - x_{i,t-1})'\beta + \gamma + \epsilon_{it} - \epsilon_{i,t-1} \tag{3}$$

which eliminates the section fixed-effect, and can be represented in matrix form as

$$FY = \rho FY_{-1} + FX\beta + \gamma 1 + F\epsilon \tag{4}$$

where

$$F = I_n \otimes F_T,$$

where

$$F_T = \begin{bmatrix} -1 & 1 & 0 & \cdots & 0 & 0 \\ 0 & -1 & 1 & \cdots & 0 & 0 \\ \vdots & & & \ddots & & \\ 0 & 0 & 0 & \cdots & -1 & 1 \end{bmatrix}_{(T-1)\times T}.$$

This first-difference operator, $F$, removes the individual effects $\alpha$ because $FD = 0$. It also inserts, on the right-hand side of the equation, a lagged differenced dependent variable $y_{i,t-1} - y_{i,t-2}$, which is correlated with the error term $\epsilon_{it} - \epsilon_{i,t-1}$, as both terms include $\epsilon_{i,t-1}$. Arellano and Bond (1991) suggested using the instrument variables $y_{i,t-2}, y_{i,t-3}, \ldots, y_{i1}$, and applied the generalized method of moments (GMM) to estimate the parameters. The instrument variables satisfy the moment condition of

$$E(y_{i,t-s}, \epsilon_{i,t} - \epsilon_{i,t-1}) = 0 \; for \; s \geq 2, t = 3, \ldots, T \tag{5}$$

The instrument variables can be represented in matrix form as

$$Z = \begin{bmatrix} Z_1 \\ \vdots \\ Z_n \end{bmatrix},$$

where $Z_i = \begin{bmatrix} y_{i1} & 0 & 0 & \cdots & 0 & 0 & \cdots & 0 \\ 0 & y_{i1} & y_{i2} & \cdots & 0 & 0 & \cdots & 0 \\ \vdots & \vdots & \vdots & \ddots & \vdots & \vdots & \cdots & \vdots \\ 0 & 0 & 0 & \cdots & y_{i1} & y_{i2} & \cdots & y_{i,T-2} \end{bmatrix}.$

Using the moment condition above, a two-step GMM estimation was performed. The first step assumes independent and homoscedastic errors across sections and over time. The residuals obtained in the first step are then used to construct a consistent variance–covariance matrix for the second step. More specifically, letting $G = [FY_{-1} \; FX \; FW]$ and $\theta = [\rho \; \beta \; \gamma]$, we have

$$\hat{\gamma}_{GMM} = (G'ZVZ'G)^{-1}(G'ZVZ'Y) \tag{6}$$

where $V = Z'F\hat{\epsilon}\hat{\epsilon}'F'Z$ and $\hat{\epsilon}$ is the residual from the first step estimation.

## 3. Case Study

*Dataset*

A case study was carried out by analyzing TxDOT's Pavement Management Information System (PMIS) data collected from 2000 to 2008 (the data from 2010–2018 was available for the analysis, but not for publication in the present study). The PMIS database contains pavement information collected from 25 districts of Texas. In PMIS, a typical pavement section is 0.8 km in length. Each section is labeled by a unique reference marker (Stampley et al. [33]). In this case study, the condition score (CS) in the PMIS database was used as the pavement performance indicator. The CS represents the pavement's overall condition in terms of both distress and ride quality, ranging from 1 (the worst condition) to 100 (the best condition). To eliminate the disruption of maintenance and rehabilitation (MR) treatments, observations with M&R interventions should be removed from the dataset. However, the history of M&R treatments was not recorded in the collected dataset. Therefore, it is not clear when an M&R treatment was applied on a section, and what treatment was used. For this reason, conditions

between consecutive years were compared. If the difference between two years' conditions ($y_{it} - y_{i,t-1}$) was greater than 1, that record was deleted.

Meteorological data was obtained from US Climate Data website. For each section in PMIS, the corresponding average temperature and precipitation values were calculated up to 23 months prior to condition inspection. The temperature and precipitation variables were denoted as *t1* to *t23* and p1 to p23, respectively. The calculated values were in the monthly format from the year 2002 to 2008. Because of the structure of the model (Equation (5)), climatic data before 2002 is not needed. During the observed period, the condition and lag variable has a maximum value of 100, a median value of 100, and a minimum value of 1; the time variable has a maximum value of 5, a median value of 3, and a minimum value of 0. In order to get a comprehensive picture for the modeling of the pavement behavior, the cumulative effect of temperature and precipitation were considered. Figures 1 and 2 shows the descriptive statistics of the variables used in this study, where "x" in the boxplot represents the cumulative average of the variable, the "– " in the middle of each boxplot represent the median of the variable, and the "−" on the top and the bottom of each boxplot represent the maximum and the minimum value of the variable respectively. The average temperature and precipitation were calculated up to 23 months prior to the condition inspection.

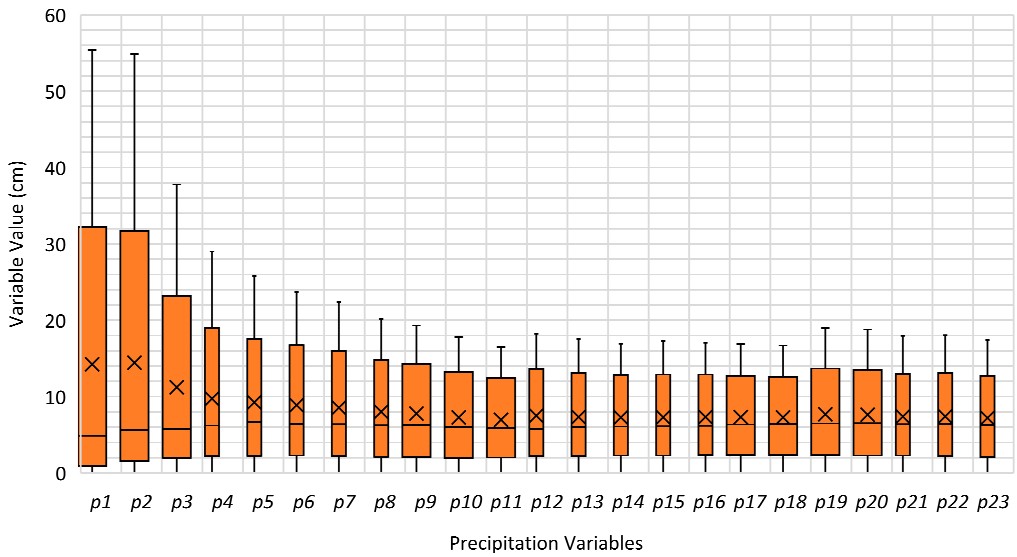

**Figure 1.** Descriptive statistics of precipitation variables.

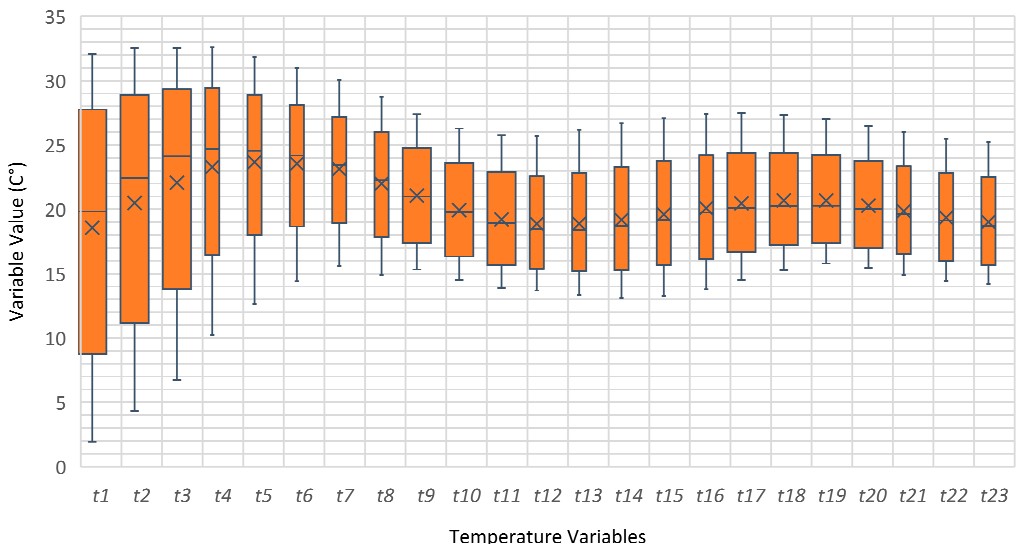

**Figure 2.** Descriptive statistics of temperature variables.

Figure 3 presents the CS deterioration rate and the average monthly rainfall from 2003 to 2008. The deterioration rate was calculated as the average CS difference between two consecutive years among all sections in the dataset. For example, the deterioration rate is 1.9 for the year 2003 in Figure 3. This means that the network-level CS value dropped around 1.9 from 2002 to 2003. The deterioration rates show an accelerating trend, as the condition dropped by 5.11 between the years 2007 and 2008. At the same time, the precipitation fluctuated between 5.99 and 8.26 cm. Figure 4 shows the trends of average temperature and deterioration rate.

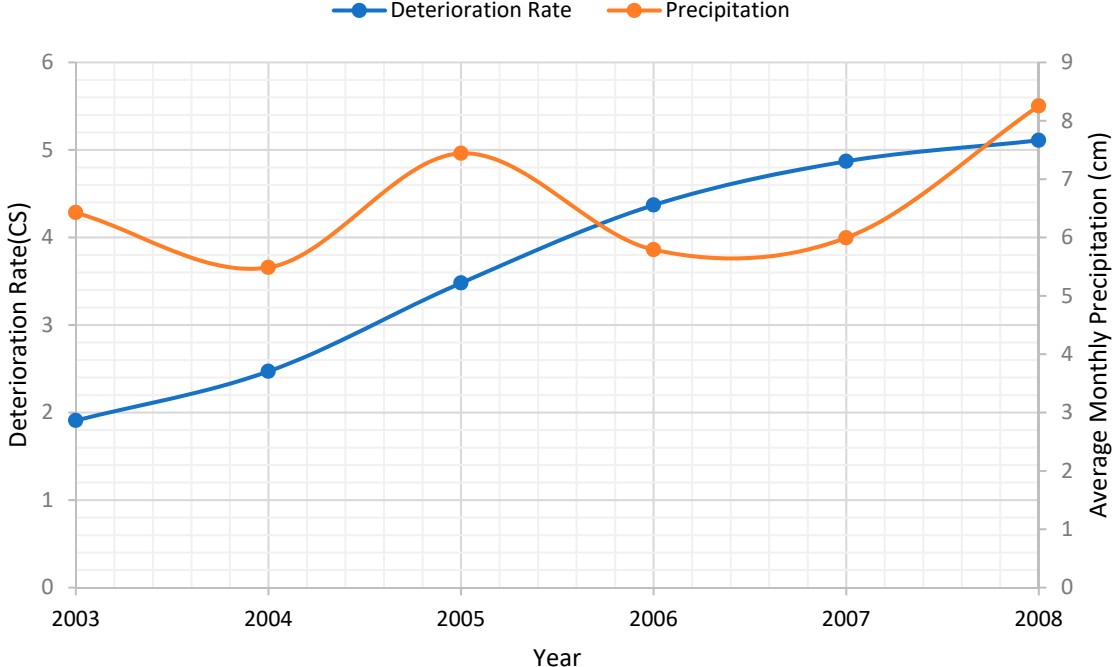

**Figure 3.** Average precipitation and deterioration rate.

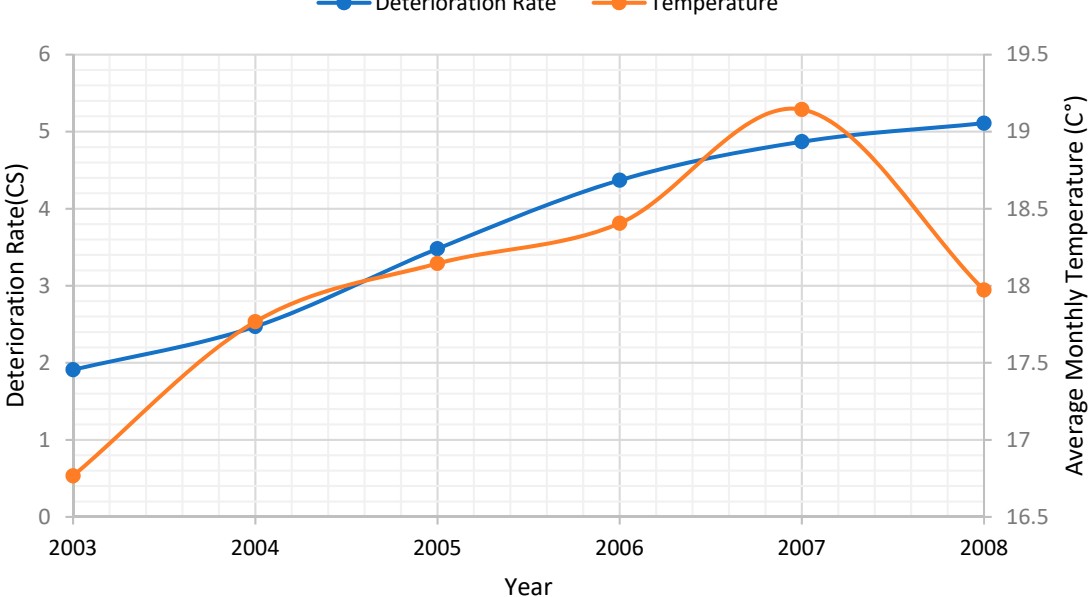

**Figure 4.** Average temperature and deterioration rate.

## 4. Estimation Results

Tables 1 and 2 present the estimation results for different model specifications. These specifications were tested to explore the effects of temperature and precipitation with different time dimensions. The numbers in the first column of the table represents different model specifications. Each specification consists of a lagged variable, a time variable, and one of the weather variables. Variables of the same specification are listed in the same row. The numbers inside the parenthesis are the *p*-values, while the numbers in the adjacent column are estimated coefficients. Significant correlations were found between the pavement condition, average monthly temperature, and the precipitation recorded 1 to 23 months prior to the condition inspection (Tables 1 and 2). Nevertheless, correlations between temperature and the condition at a lag of 5 and 7 months are not statistically significant at the 0.05 level. A positive sign of the coefficient indicates that the variable contributes to the condition improvement, whereas a negative sign indicates that the variable contributes to the deterioration. For example, the negative coefficients of all the precipitation variables indicate that rainfall is always detrimental to pavement conditions, based on the estimation. The coefficient of the lagged variable (last year's condition) is significant at all model specifications, which indicates the dynamic process of pavement deterioration.

**Table 1.** Estimation results for time, lag performance, and temperature.

| Specification | Time Variable | | Lag Variable | | Weather Variable | | |
|---|---|---|---|---|---|---|---|
| 1 | −1.5175 | (0.0) | 0.7043 | (0.0) | $t1$ | 0.0265 | (0.0) |
| 2 | −1.5185 | (0.0) | 0.7042 | (0.0) | $t2$ | 0.0241 | (0.0002) |
| 3 | −1.516 | (0.0) | 0.7042 | (0.0) | $t3$ | 0.0208 | (0.0038) |
| 4 | −1.5149 | (0.0) | 0.7041 | (0.0) | $t4$ | 0.0198 | (0.0246) |
| 5 | −1.5127 | (0.0) | 0.704 | (0.0) | $t5$ | 0.0192 | (0.0831) |
| 6 | −1.5073 | (0.0) | 0.704 | (0.0) | $t6$ | 0.0064 | (0.3685) |
| 7 | −1.4978 | (0.0) | 0.7042 | (0.0) | $t7$ | -0.0362 | (0.0851) |
| 8 | −1.4852 | (0.0) | 0.7046 | (0.0) | $t8$ | -0.0831 | (0.0025) |
| 9 | −1.4828 | (0.0) | 0.7047 | (0.0) | $t9$ | −0.1071 | (0.0001) |
| 10 | −1.4935 | (0.0) | 0.7045 | (0.0) | $t10$ | −0.077 | (0.0046) |
| 11 | −1.4946 | (0.0) | 0.7045 | (0.0) | $t11$ | −0.0638 | (0.0293) |
| 12 | −1.5074 | (0.0) | 0.704 | (0.0) | $t12$ | 0.01 | (0.403) |
| 13 | −1.5199 | (0.0) | 0.7037 | (0.0) | $t13$ | 0.0778 | (0.0221) |
| 14 | −1.5181 | (0.0) | 0.7038 | (0.0) | $t14$ | 0.0687 | (0.0241) |
| 15 | −1.5151 | (0.0) | 0.704 | (0.0) | $t15$ | 0.0448 | (0.0853) |
| 16 | −1.51 | (0.0) | 0.704 | (0.0) | $t16$ | 0.018 | (0.2966) |
| 17 | −1.5029 | (0.0) | 0.704 | (0.0) | $t17$ | −0.011 | (0.3871) |
| 18 | −1.4872 | (0.0) | 0.704 | (0.0) | $t18$ | −0.0763 | (0.049) |
| 19 | −1.459 | (0.0) | 0.7041 | (0.0) | $t19$ | −0.1878 | (0.0003) |
| 20 | −1.4262 | (0.0) | 0.7042 | (0.0) | $t20$ | −0.2985 | (0.0) |
| 21 | −1.407 | (0.0) | 0.7043 | (0.0) | $t21$ | −0.3542 | (0.0) |
| 22 | −1.3925 | (0.0) | 0.7045 | (0.0) | $t22$ | −0.3977 | (0.0) |
| 23 | −1.3779 | (0.0) | 0.7042 | (0.0) | $t23$ | −0.4679 | (0.0) |

The data from 2010–2018 was available for the analysis, but not for publication in the present study.

**Table 2.** Estimation results for time, lag performance, and precipitation.

| Specification | Time Variable | | Lag Variable | | Weather Variable | | |
|---|---|---|---|---|---|---|---|
| 1 | −1.5048 | (0.0) | 0.7031 | (0.0) | $p1$ | −0.0946 | (0.0) |
| 2 | −1.4956 | (0.0) | 0.7031 | (0.0) | $p2$ | −0.1082 | (0.0) |
| 3 | −1.4951 | (0.0) | 0.703 | (0.0) | $p3$ | −0.1149 | (0.0) |
| 4 | −1.4952 | (0.0) | 0.7025 | (0.0) | $p4$ | −0.1293 | (0.0) |
| 5 | −1.4828 | (0.0) | 0.7022 | (0.0) | $p5$ | −0.1562 | (0.0) |
| 6 | −1.4691 | (0.0) | 0.701 | (0.0) | $p6$ | −0.2163 | (0.0) |
| 7 | −1.4637 | (0.0) | 0.7014 | (0.0) | $p7$ | −0.1966 | (0.0) |
| 8 | −1.4616 | (0.0) | 0.7014 | (0.0) | $p8$ | −0.2218 | (0.0) |
| 9 | −1.4653 | (0.0) | 0.7014 | (0.0) | $p9$ | −0.2203 | (0.0) |
| 10 | −1.4644 | (0.0) | 0.7012 | (0.0) | $p10$ | −0.2545 | (0.0) |
| 11 | −1.4622 | (0.0) | 0.7008 | (0.0) | $p11$ | −0.2895 | (0.0) |
| 12 | −1.4718 | (0.0) | 0.7002 | (0.0) | $p12$ | −0.2983 | (0.0) |
| 13 | −1.4736 | (0.0) | 0.7 | (0.0) | $p13$ | −0.3019 | (0.0) |
| 14 | −1.4693 | (0.0) | 0.6999 | (0.0) | $p14$ | −0.3271 | (0.0) |
| 15 | −1.4776 | (0.0) | 0.7009 | (0.0) | $p15$ | −0.2875 | (0.0) |
| 16 | −1.4894 | (0.0) | 0.7018 | (0.0) | $p16$ | −0.2279 | (0.0) |
| 17 | −1.4936 | (0.0) | 0.7021 | (0.0) | $p17$ | −0.2159 | (0.0002) |
| 18 | −1.4959 | (0.0) | 0.7025 | (0.0) | $p18$ | −0.1914 | (0.0016) |
| 19 | −1.4991 | (0.0) | 0.703 | (0.0) | $p19$ | −0.1551 | (0.0103) |
| 20 | −1.4998 | (0.0) | 0.703 | (0.0) | $p20$ | −0.1637 | (0.0085) |
| 21 | −1.4983 | (0.0) | 0.7029 | (0.0) | $p21$ | −0.1797 | (0.0056) |
| 22 | −1.4997 | (0.0) | 0.7028 | (0.0) | $p22$ | −0.2011 | (0.0035) |
| 23 | −1.5019 | (0.0) | 0.703 | (0.0) | $p23$ | −0.197 | (0.0089) |

The data from 2010–2018 was available for the analysis, but not for publication in the present study.

Figures 5 and 6 show the effects of temperature and precipitation based on the results of Tables 1 and 2. These effects represent a unit change in network level pavement condition score, due to a unit change in the contributing variable (either temperature or precipitation). In formulation, these effects correspond to vector $\beta$. However, only those coefficients whose *p*-values were smaller than 0.05 were displayed. As shown in Figure 5, the effects of temperature on pavement condition switched between positive and negative values over time. For example, the average temperatures within five months prior to the inspection have a positive effect on the pavement condition value. However, the magnitudes of these effects are small. The effects of temperature between 5 months and 12 months prior to the inspection are negative. Then the effects become positive again between 12 months and 16 months. The long-term effects of temperature (16+ months prior to the inspection) are negative, and the magnitudes are significant. Figure 6 shows the effect of precipitation on the condition of pavement. This figure suggests that precipitation has a negative impact on the condition of the pavement at all time dimensions. However, average precipitation within one year prior to the inspection has the greatest impact on the condition.

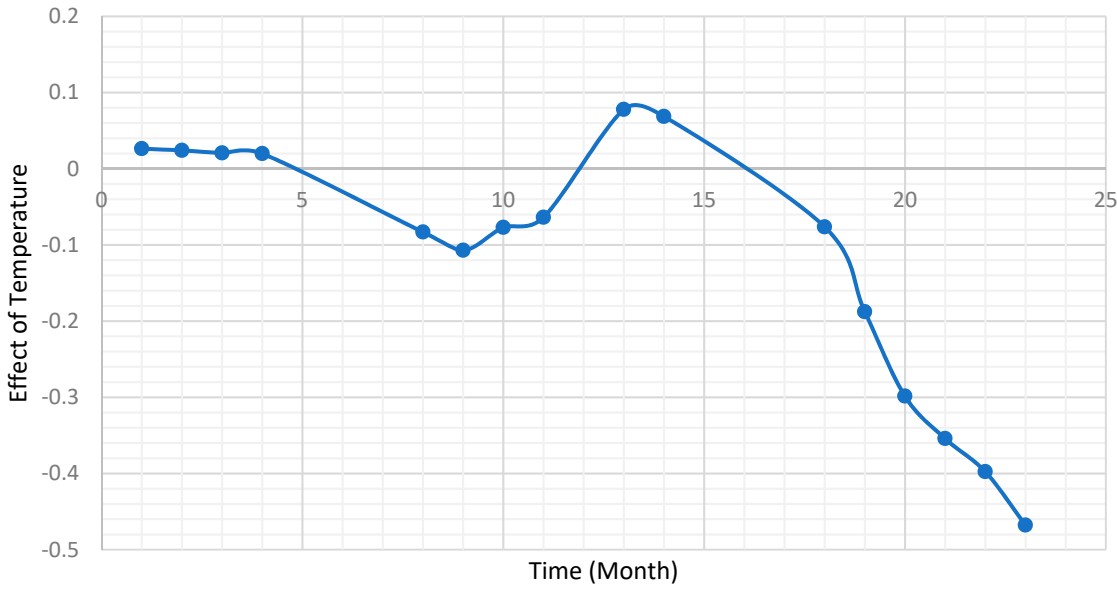

**Figure 5.** Effects of temperature.

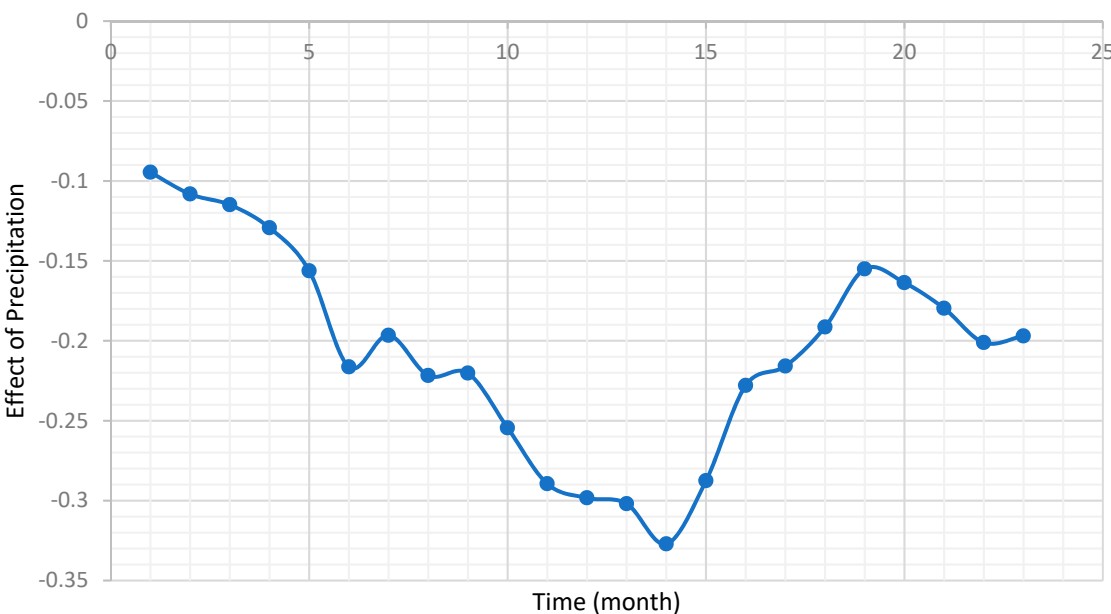

**Figure 6.** Effects of precipitation.

## 5. Conclusions

The study analyzed the effects of temperature and precipitation variations on pavement performance using a dynamic panel model. Data from the Texas road network collected between 2000 and 2008 was used for the estimation. Average temperature and precipitation from 1 to 23 months prior to the condition inspection were used in the analysis. The results demonstrate that average monthly rainfall and temperature have significant effects on pavement condition. While the short-term effects of temperature on pavement condition fluctuates between positive and negative over time, the long-term effects of temperature (16+ months prior to the inspection) are negative and the magnitudes are significant. This indicates that as the global warming is exacerbated, pavements designed based on historical data will suffer from a faster deterioration rate. The results also show that precipitation always has negative impact on the condition of the pavement at all time dimensions, and the maximum effect of precipitation was found to be the average monthly rainfalls over the single year prior to the inspection. Moreover, it was found the lagged pavement condition variable is significant at all

model specifications. The deterioration rate will be significantly affected by both the precipitation and temperature.

In summary, these findings could facilitate (1) evaluating network-level pavement condition data when inspections are conducted at different seasons, and (2) understanding network-level pavement deterioration patterns by taking both short-term (seasonal) and annual weather variations into consideration.

**Author Contributions:** Conceptualization, L.G., F.H., Y.-H.R.; Methodology, L.G., F.H., and Y.-H.R.; Writing, review and editing: L.G., F.H., and Y.-H.R.

**Funding:** This research received no external funding.

**Acknowledgments:** The authors would like to acknowledge the resources provided by the Department of Construction Management at University of Houston and made this research. possible.

**Conflicts of Interest:** The authors declare no conflict of interest.

## Nomenclature

| Variables | Explanation |
| --- | --- |
| $i$ | Index for pavement sections |
| $t$ | Index for inspection time periods |
| $y_{it}$ | The $i$th pavement section's condition at inspection time period $t$ |
| $x_{it}$ | Covariates of the $i$th pavement section at inspection time period $t$ |
| $\alpha_i$ | Unobserved specific factor of the $i$th pavement section |
| $\epsilon_{it}$ | Error term |
| $\beta$, $\rho$ and $\gamma$ | Regression Coefficients |
| $Y$ | Vector of $y_{it}$ values |
| $X$ | Vector of $x_{it}$ values |
| $\beta$ | Vector of $\beta$ values |
| $W$ | Vector of $t$ values |
| $D$ | Block diagonal matrix |
| $Z$ | Instrument Variables Matrix |
| $F$ | First-difference operator |

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
