# Peer review of "Impacts of Seasonal and Annual Weather Variations on Network-Level Pavement Performance"

_infrastructures, doi:10.3390/infrastructures4020027_

Round 1

Reviewer 1 Report

The literature review is extended and well done. The topic of the research is well presented.

Hereinafter are reported some observations for the authors:

- @line 73 the AASHO acronym is explained but it is not the first time that it has been used, I suggest to use the extended version @line 67

 -In the methodology part the parameter beta could be better explained, and the full methodology could be better presented with minor reviews.

- The matrices @lines 107 and 108 some have the dimension subscripted while some others don't, pay attention to this detail
- Table 1: could it be restricted in the page margins?

- Table 2: could it be restricted in two pages?

- Figure 3: specify in the title of the y-axis that is the effect of the temperature, as it has been done in Figure 4 for the effect of precipitation.

Is it possible to specify the observed months, like january, february etc... to have better perception of the weather conditions during that period of the year?

Author Response

1.     @line 73 the AASHO acronym is explained but it is not the first time that it has been used, I suggest to use the extended version @line 67

Response: We thank the reviewer for this comment. Extended version has been used on line 67.

2.     In the methodology part the parameter beta could be better explained, and the full methodology could be better presented with minor reviews.

Response: We thank the reviewer for this comment.  is the coefficient of the covariates. This explanation has been added to line 133.

3.     The matrices @lines 107 and 108 some have the dimension subscripted while some others don't, pay attention to this detail

Response: We thank the reviewer for this comment. Dimension has been added to all matrices.

4.     Table 1: could it be restricted in the page margins?

Response: We thank the reviewer for this comment. Table 1 has been reformatted and restricted in the page margins.

5.     Table 2: could it be restricted in two pages?

Response: We thank the reviewer for this comment. Table 2 has been reformatted and restricted in the page margins.

6.     Figure 3: specify in the title of the y-axis that is the effect of the temperature, as it has been done in Figure 4 for the effect of precipitation.

Response: We thank the reviewer for this comment. The y-axis has been changed as suggested.

7.     Is it possible to specify the observed months, like january, february etc... to have better perception of the weather conditions during that period of the year?

Response: We thank the reviewer for this comment. The month in this research stands for the number of months before inspection date. Since the research has been conducted on different road section across Texas, therefore, the inspection date would be different. As a result, the weather condition data would be different among different road sections as well. 

Reviewer 2 Report

Authors should upgrade their study by enriching the results interpretation as well as their methodology. Although it is on a good base it does not reach a good level for being published in the present form. 

Some comments:

Tables can not offer cohesive information to readers, table 1 may be more useful to be in the form of box plots. Table 2  cannot be in a published paper in this form. Maybe some results can be presented from table 2 but in this form, it does not offer any crucial information to readers. Regarding figures 3-4, some comments on the precipitation and temperature trends should be given. Also in Figures 3-4 the term "effect" has to be identified. Then, references should be properly included as some are missing eg Loizos and Karlaftis 2005. 

The most important however is that authors should prove using other data that their methodology does work and it is not applicable only for the current set of data.

Author Response

1.     table 1 may be more useful to be in the form of box plots.

Response: Thank you Sir/Madam for your comment

Table 1 has been changed to box-plots

2.     Table 2  cannot be in a published paper in this form. Maybe some results can be presented from table 2 but in this form, it does not offer any crucial information to readers. 

Response: Thank you Sir/Madam for your comment

Table 2 has been reformatted

3.     Regarding figures 3-4, some comments on the precipitation and temperature trends should be given. Also, in Figures 3-4 the term "effect" has to be identified. 

Response: Thank you Sir/Madam for your comment

The average yearly temperature from 2000 to 2008 range from 65.3  to 69.3 where 2002 has the lowest temperature of 65.3  and 2006 has the highest temperature of 69.3. However, since the research is conducted on different road sections across Texas, therefore, the inspection date for each road section would be different. As a result of that, 1 month or 2 months before inspection date for each road section would also be different, and their temperature and precipitation trend would also be different.

4.     Then, references should be properly included as some are missing eg Loizos and Karlaftis 2005. 

Response: Thank you Sir/Madam for your comment

Reference has been added to reference list at the end of the paper

5.     The most important however is that authors should prove using other data that their methodology does work, and it is not applicable only for the current set of data.

Response: Thank you Sir/Madam for your comment

An extra paragraph has been added to the beginning of Methodology section to show that the dynamic panel model has been applied to different data set and modified for different purpose in various researches. 

Round 2

Reviewer 2 Report

-Authors should avoid using the 1st plural as in line 102.

-Tables 1-2  should be improved as they are not visible in such form.

-The color of box plots should change so as the points inside to become visible.

-Why there is such fluctuation among the first and last box-plots (height of box plots)? How the authors interpret this? 

-Conclusions should be enhanced with some comments on the findings i.e. how temperature changes affect the pavement performance?

-Are there previous efforts that are on the same track with the authors' findings, except those presented in the introduction section. Mainly analyzing the same variables as affecting factors in pavement long-term performance.

Author Response

1.     Authors should avoid using the 1st plural as in line 102.

Response: Thank you Sir/Madam for your comment

All first-person plural has been rephrased.

2.     Tables 1-2 should be improved as they are not visible in such form.

Response: Thank you Sir/Madam for your comment

Table 1-2 has been reformatted.

3.     The color of box plots should change so as the points inside to become visible.

Response: Thank you Sir/Madam for your comment

The color of the box plots has been changed.

4.     Why there is such fluctuation among the first and last box-plots (height of box plots)? How the authors interpret this? 

Response: Thank you Sir/Madam for your comment

The previous fluctuation is caused by plotting condition, lag, and time variable on the same box plot. The condition and lag variable have been removed from figure to eliminate the unnecessary fluctuation, and statistics of these two variables have been moved to line 209.

5.     Conclusions should be enhanced with some comments on the findings i.e. how temperature changes affect the pavement performance?

Response: Thank you Sir/Madam for your comment

Additional comments on the effects of temperature and precipitation have been added to the conclusion section at line 247.

During the 24-month observation period, both the temperature and precipitation has an overall negative effect on pavement condition. Figure 5 shows a significant negative effect on pavement performance at the end of the observation period. This indicates that, in the long-term, as the pavement experience more variation in temperature, the deterioration rate of pavement will accelerate; and as the global warming exasperates, pavements designed based on historical data will suffer from a faster deterioration rate. On the other hand, figure 6 indicates that as the pavements experience more precipitation, the pavement deterioration rate will also accelerate. Combining figure 5 and figure 6 together, the study shows that pavement deterioration rate will be significantly affected by both the precipitation and temperature since the accumulated precipitation and varying temperature will cause thawing and freezing of the moisture content inside the porous structure of pavement sections, and also causing deformation of the underlying soil.

 6.     Are there previous efforts that are on the same track with the authors' findings, except those presented in the introduction section. Mainly analyzing the same variables as affecting factors in pavement long-term performance.

Response: Thank you Sir/Madam for your comment

The following literature reviews have been added to the introduction section to demonstrate previous efforts on the same track. Mainly analyzing the same variables as affecting factors in pavement long-term performance.

        1.     Mohamed, A. R., & Hansen, W. (1997). Effect of nonlinear temperature gradient on curling         stress in concrete pavements. Transportation Research Record, 1568(1), 65-71.

        Mohamed and Hansen (1997) used wheel loading temperature and moisture gradients as the         affecting factors to study the tensile stress at both the top and bottom of a slab caused by. The         first part of the study presented the calculation of self-equilibrated stresses caused by internal         restraint within a cross-section. The second part of the study calculated stresses caused by             external restraint using an equivalent linear temperature gradient provided by the first part of         the study.

        2.     Bosscher, P. J., Bahia, H. U., Thomas, S., & Russell, J. S. (1998). Relationship             between pavement temperature and weather data: Wisconsin field study to verify the         superpave algorithm. Transportation Research Record, 1609(1), 1-11.

        Bosscher et al (1998) developed a statistical model to analyze the pavement temperature             using meteorological data. The study used air temperature, relative humidity, wind speed and         direction, and solar radiation as the affecting variable. The developed model was compared             with superpave model and model recommended by the LTPP program. The results of this             study indicate that at air temperature higher than, the high pavement design temperature was         underestimated by the superpave model and LTPP model. The study also found a significant         difference between the standard deviation of air and pavement temperature.

        3.     Yu, H. T., Khazanovich, L., Darter, M. I., & Ardani, A. (1998). Analysis of                 concrete pavement responses to temperature and wheel loads measured from                     instrumented slabs. Transportation Research Record, 1639(1), 94-101.

        Yu et al (1998) utilized both temperature and wheel loads as the affecting variable and studied         concrete pavement responses to the affecting variables. It was discovered that the built-in             upward curling of a concrete slab shares a similar effect as the negative temperature                     gradients.

        4.     Bäckström, M. (2000). Ground temperature in porous pavement during freezing             and thawing. Journal of transportation engineering, 126(5), 375-381.

        Bäckström (2000) utilized both temperature and precipitation as the affecting variable, and it         was found that a porous pavement is more resistant to freezing than traditional impermeable         pavement due to higher latent ground heat caused by higher water composition in the                     underlying soil.

        5.     Marasteanu, M. O., Li, X., Clyne, T. R., Voller, V., Timm, D. H., & Newcomb, D. (2004).             Low temperature cracking of asphalt concrete pavement.

        In marasteanu et al (2004) two models were developed to study the cracking mechanism of             asphalt pavement exposed to low temperature. The first model used temperature as the                 affecting variable to predict the crack spacing in asphalt pavement exposed to low                         temperature. The second model also used temperature as the affecting variable to predict             cracking propagation and accumulated damage. The result shows that after a cracking start, it         only takes around decrease in temperature for the crack to propagate.

        6.     Bazlamit, S. M., & Reza, F. (2005). Changes in asphalt pavement friction                     components and adjustment of skid number for temperature. Journal of Transportation         Engineering, 131(6), 470-476.

        Bazlamit and Reza (2005) temperature and surface texture were used as the affecting                     variable to study the friction force developed during skidding. Ten field sites with different                 pavement types and five different temperatures were used in this study. The result shows that         the hysteresis component of friction decreases as temperature increase regardless of the             pavement texture state. The adhesion component was more sensitive to the effect of surface         texture.

        7.     Qiao, Y., Flintsch, G. W., Dawson, A. R., & Parry, T. (2013). Examining effects of         climatic factors on flexible pavement performance and service life. Transportation             Research Record, 2349(1), 100-107.

        In qiao et al (2013), climate factors such as, temperature variation, precipitation, wind speed,         and groundwater level were used as affecting variables to examine pavement deterioration             and service life through sensitivity analysis. It was concluded that both annual temperatures         increase and seasonal temperature variation has the most significant effect on pavement                 deterioration rate and pavement service life.

        8.     Mohd Hasan, M. R., Hiller, J. E., & You, Z. (2016). Effects of mean annual temperature and         mean annual precipitation on the performance of flexible pavement using ME                                 design. International Journal of Pavement Engineering, 17(7), 647-658.

        Mohd Hasan et al (2016) used mean temperature and precipitation as the affecting factors to         study the effect on the distress of flexible pavement. Seventy-six pavement section from                 thirteen states with different weather condition were utilized in this study. The results show that         longitudinal cracking of flexible pavement was significantly affected by both temperature and         precipitation, whereas alligator cracking, transverse cracking on permanent deformation were         only affected significantly by mean annual temperature. However, both temperature and                 precipitation did not show a significant effect on the international roughness of flexible                     pavement.
